# Fractal Derivatives, Fractional Derivatives and *q*-Deformed Calculus

**DOI:** 10.3390/e25071008

**Published:** 2023-06-30

**Authors:** Airton Deppman, Eugenio Megías, Roman Pasechnik

**Affiliations:** 1Instituto de Física, Universidade de São Paulo, São Paulo 05508-090, Brazil; 2Departamento de Física Atómica, Molecular y Nuclear and Instituto Carlos I de Física Teórica y Computacional, Universidad de Granada, Avenida de Fuente Nueva s/n, 18071 Granada, Spain; emegias@ugr.es; 3Department of Physics, Lund University, Sölvegatan 14A, SE-22362 Lund, Sweden; roman.pasechnik@hep.lu.se

**Keywords:** fractal derivatives, fractional derivatives, fractional differential equations, *q*-calculus, nonextensive statistics

## Abstract

This work presents an analysis of fractional derivatives and fractal derivatives, discussing their differences and similarities. The fractal derivative is closely connected to Haussdorff’s concepts of fractional dimension geometry. The paper distinguishes between the derivative of a function on a fractal domain and the derivative of a fractal function, where the image is a fractal space. Different continuous approximations for the fractal derivative are discussed, and it is shown that the *q*-calculus derivative is a continuous approximation of the fractal derivative of a fractal function. A similar version can be obtained for the derivative of a function on a fractal space. Caputo’s derivative is also proportional to a continuous approximation of the fractal derivative, and the corresponding approximation of the derivative of a fractional function leads to a Caputo-like derivative. This work has implications for studies of fractional differential equations, anomalous diffusion, information and epidemic spread in fractal systems, and fractal geometry.

## 1. Introduction

Fractional differential equations have been used to describe the behavior of complex systems. The growing interest in this mathematical tool imposes the necessity of urgent analysis of its fundamentals. The widespread use of fractional differential equations in fluid dynamics, finance, and other complex systems has led to the intense investigation of the properties of fractional derivatives and their geometrical and physical meaning. Fractional derivatives are often associated with fractal geometry, but the connections between fractional derivatives and fractal derivatives have not been clarified so far. Fractional derivatives have been used in many applications [1,2], and advancing our understanding of their geometrical meaning and their relations with fractals is necessary. The *q*-calculus has been frequently applied to describe the statistical properties of fractal systems [3,4]. However, the relationship between *q*-calculus and fractal derivatives has not been fully understood yet.

This work reviews the fundamentals of fractal derivatives and establishes their connections with fractional derivatives and *q*-calculus. The generalization of standard calculus to include fractional-order derivatives and integrals is an exciting field of research, and many works have been conducted in this area. Different proposals for fractional generalization are available, and applications of fractional derivatives have been used in various fields. Fractional differential equations are frequently used to describe the behavior of complex systems. In Refs. [5,6], the authors analyzed different forms of fractional derivatives and discussed their properties. Caputo’s derivative is among the most commonly used and is defined by
(1)$$D_{C}^{\nu }h\left(x\right)=\frac{1}{\Gamma (1-\nu )}\int_{x-\delta }^{x}{\left(x-t\right)}^{-\nu }\frac{dh}{dt}dt\;,$$
which is a particular case of the Riemann–Liouville fractional derivative [7].

Haussdorff established the fundamental aspects of spaces with fractional dimension, and an introduction to the subject can be found in [8]. One of the important quantities associated with fractal spaces is the Haussdorff measure, denoted by $H^{s}\left(F\right)$. Its definition is based on the measure $H_{\delta }^{s}\left(F\right)$, and is given by
(2)$$H^{s}\left(F\right)=\lim_{\delta \to 0}H_{\delta }^{s}\left(F\right)\;,$$
where the measure depends on a δ-cover of the Borel subset $F\subseteq R^{n}$. The space F will be referred to as a fractal space, and its Hausdorff dimension is denoted by α and defined as
(3)$$\alpha =\inf\left\{s\geq 0:H^{s}\left(F\right)=0\right\}=\sup\left\{s:H^{s}\left(F\right)=\infty \right\}\;.$$

If 0<α<∞, the Haussdorff measure of the $\delta^{\alpha }$-cover is called the mass distribution, denoted by $\gamma^{\alpha }\left(F,a,b\right)$ [9,10,11], which will be discussed below. Fractal derivatives and fractional derivatives are not the same concept [12], and the non-locality is a prominent aspect of the fractal derivative. For a comprehensive review of the subject and its applications, see Ref. [13]. The Parvate–Gangal derivative is defined for functions on a fractal domain. This work shows that extending the same concepts to functions with a fractal image can provide new insights into the role of fractal derivatives in the study of complex systems.

Tsallis statistics was proposed to describe the statistical properties of fractal systems. It introduces a non-additive entropy that can be used to obtain, through the ordinary thermodynamics formalism, the non-extensive thermodynamics [14,15]. To deal with non-additivity, the *q*-calculus was proposed [16]. One important result of *q*-calculus is the *q*-derivative, which is written as:(4)$$\frac{\bar{d}f}{dx}=f^{q-1}\frac{df}{dx}.$$
Notice that, if the function *f* is a *q*-exponential, the *special derivative* above results to be identical to the standard derivative of a *q*-exponential function. This derivative can be straightforwardly related to the conformal derivative [17].

The three different theoretical areas mentioned above have been investigated independently, evolving in parallel. Despite their many common aspects, the connections between them have not been demonstrated so far [18]. This work aims to establish connections between Caputo’s derivative and the *q*-calculus with the continuous approximation of the fractal derivative proposed by Parvate and Gangal. In this work, we assume that the fractal derivative is correctly calculated by the definitions advanced by Parvate, Gangal, and coworkers [9,10,11], and discuss how some relevant forms of fractional derivatives, as well as the *q*-deformed derivative, can be obtained as a continuous approximation of the fractal derivative.

## 2. Fractal Derivatives

**Lemma 1.** 
*If $x=\left(x_{1},\cdots ,x_{n}\right)\in R^{n}$ and $f=\left(f_{1}\left(x\right),\cdots ,f_{m}\left(x\right)\right)\in R^{m}$ is an m-dimensional vector field $f:R^{n}\to R^{m}$. Then, m≤n.*


**Proof.** Suppose m>n, then $\left(f_{1}\left(x\right),\cdots ,f_{n}\left(x\right)\right)$ forms a new set of *n* independent variables, which are functions of the *n* independent variables of *x*. Then, $f_{n+1}\left(x\right)$ is not independent of the functions in the set. □

**Definition 1.** 
*A vector field with dimension m=1 is a function.*


**Lemma 2.** 
*If there is an inverse function $f^{-1}\left(f(x)\right)=x$, then m=n.*


**Proof.** It follows immediately by applying Lemma 1. □

**Lemma 3.** 
*If f is a fractal vector field $f:R^{n}\to R^{\alpha }$, with α∈R, then α≤n.*


**Proof.** It follows immediately by applying Lemma 1. □

**Definition 2.** 
*A fractal vector field with dimension α≤1 is a fractal function.*


**Definition 3.** 
*An α-dimensional function is a fractal vector field if α>1 or a fractal function if α≤1.*


**Definition 4.** 
*If γ(F,a,b) is the Haussdorff mass distribution for a cover F, with a,b∈F, then the staircase function, $S_{F,a_{o}}^{\alpha }$, is defined as*

(5)
$$S_{F,a_{o}}^{\alpha }=\left\{\begin{array}{c} \gamma \left(F,a_{o},x\right)\;\;for\;x>a_{o}\\ \gamma \left(F,x,a_{o}\right)\;\;for\;x<a_{o} \end{array}\right.\;.$$



**Lemma 4.** 
*The staircase function is a scalar.*


**Proof.** The staircase function is proportional to the Haussdorff mass function, which is the volume resulting from the union of the $\delta^{\alpha }\left(x\right)\in F$, so it is a scalar. □

**Definition 5.** 
*If F is a $\delta^{\alpha }$-cover and f:F→R, then the fractal derivative of f(x) is*

(6)
$$D_{F,a_{o}}^{\alpha }f\left(x_{o}\right)=\left\{\begin{array}{c} F\lim_{x\to x_{o}}\frac{f\left(x\right)-f\left(x_{o}\right)}{S_{F,a_{o}}^{\alpha }\left(x\right)-S_{F,a_{o}}^{\alpha }\left(x_{o}\right)}\;\;\;\;x,x_{o}\in F\\ 0\;\;\;\;otherwise \end{array}\right.\;.$$



**Theorem 1.** 
*There is a fractal derivative of the inverse function, and it is the inverse of the fractal derivative.*


**Proof.** Consider that $x,x_{o}\in F$. Suppose there exists a function g:R→F such that g(f(x))=x. Then,
(7)$$D_{F,a_{o}}^{\alpha }g\left(f_{x_{o}}\right)=F\lim_{x\to x_{o}}\frac{g\left(f_{x}\right)-g\left(f_{x_{o}}\right)}{f\left(x\right)-f\left(x_{o}\right)}\frac{f\left(x\right)-f\left(x_{o}\right)}{S_{F^{\prime },a_{o}}^{\alpha }\left(x\right)-S_{F^{\prime },a_{o}}^{\alpha }\left(x_{o}\right)}=1\;,$$
where the simplified notation $f_{x}=f\left(x\right)$ was adopted. It follows that
(8)$$F\lim_{x\to x_{o}}\frac{g\left(f_{x}\right)-g\left(f_{x_{o}}\right)}{f\left(x\right)-f\left(x_{o}\right)}=F\lim_{x\to x_{o}}\frac{S_{F^{\prime },a_{o}}^{\alpha }\left(x\right)-S_{F^{\prime },a_{o}}^{\alpha }\left(x_{o}\right)}{f\left(x\right)-f\left(x_{o}\right)}\;.$$□

The fractal derivative of the inverse function can be applied to any fractal function *h*: R→F. The staircase function, in this case, is applied to the fractal image space of the function *h*. The function *f* can be defined arbitrarily, with the constraint that there is an inverse function $f^{-1}$. One case of particular interest is the identity function f(x)=x, then we have
(9)$${\left[D_{F,\varphi }^{\alpha }\right]}^{-1}h\left(x_{o}\right)=F\lim_{x\to x_{o}}\frac{S_{F,\varphi }^{\alpha }\left[h\left(x\right)\right]-S_{F,\varphi }^{\alpha }\left[h\left(x_{o}\right)\right]}{x-x_{o}}\;,$$
with $\varphi =h(a_{o})$.

Observe that in this case, the image space and the domain space of the function *h* are the same, i.e., *h*: F→F.

**Definition 6.** 
*The result obtained above can be generalized by defining the fractal derivative of the inverse function or, equivalently, the inverse of the fractal derivative, as*

(10)
$${\left[D_{F,\varphi }^{\alpha }\right]}^{-1}h\left(f_{x_{o}}\right)=\left\{\begin{array}{c} F\lim_{x\to x_{o}}\frac{S_{F,\varphi }^{\alpha }\left[h\left(x\right)\right]-S_{F,\varphi }^{\alpha }\left[h\left(x_{o}\right)\right]}{x-x_{o}}\;\;\;\;x,x_{o}\in F\;.\\ 0\;otherwise \end{array}\right.$$



**Corollary 1.** 
*The derivative of a fractal function is well-defined only if the function is almost always non-divergent in the interval [a,b] (Following the standard terminology in the field, we say that a function is almost always non-divergent if the set of points where it is divergent has null Lebesgue measure).*


**Proof.** According to Definition 4, the staircase function is well-defined only if the mass distribution function can be defined. The mass distribution is equal to the Haussdorff measure when the Haussdorff dimension is 0<α<∞. This condition is satisfied only if the function is almost always non-divergent. □

**Theorem 2.** 
*If the function h(x) is almost always continuous and non-divergent in F, and $h^{\prime }\left(x\right)={\left[D_{F,\varphi }^{\alpha }\right]}^{-1}h\left(x\right)$, then the Haussdorff dimension of h(x) and $h^{\prime }\left(x\right)$ are the same.*


**Proof.** Let F be the $\delta^{\alpha }$-cover of the fractal function h(x), and $F^{\prime }$ the $\delta^{\beta }$-cover of the inverse of fractal derivative. For any $\delta^{\alpha }\left[h\left(x\right)\right]\in F$ there is a $\delta^{\beta }\left[h^{\prime }\left(x\right)\right]\in F^{\prime }$, so β≥α. For $\delta^{\beta }\left[h^{\prime }\left(x\right)\right]\in F^{\prime }$, there is a $\delta^{\alpha }\left[h\left(x\right)\right]\in F$; therefore, α≤β. Hence, α=β. □

**Definition 7.** 
*We will denote the inverse of an α-dimensional fractal function by $D_{F,\varphi }^{\alpha }h\left(x\right)$, and we will refer to it as a fractal derivative of an α-dimensional fractal function, or simply fractal function, while the fractal derivative will be called fractal derivative over a fractal space.*


**Definition 8.** 
*The partial derivative of a fractal function is*

(11)
$$D_{F,\varphi }^{\alpha }{|}_{i}h\left(f_{x}\right)=\left\{\begin{array}{c} F\lim_{x_{i}\to x_{o,i}}\frac{S_{F,\varphi }^{\alpha }\left[h\left(x\right)\right]-S_{F,\varphi }^{\alpha }\left[h\left(x_{o}\right)\right]}{x_{i}-x_{o,i}}\;\;\;\;x,x_{o}\in F\;,\\ 0\;otherwise \end{array}\right.$$

*where the index i indicates the component $x_{i}$ of the vector x.*


**Corollary 2.** 
*The dimension of $D_{F,\varphi }^{\alpha }{|}_{i}\;h\left(f_{x}\right)$ is α≤1.*


**Proof.** It follows immediately from Lemma 1 and Theorem 2. □

**Definition 9.** 
*The staircase function differential is defined by*

(12)
$$dS_{F,a_{o}}^{\alpha }\left(x\right)=\left\{\begin{array}{c} F\lim_{dx\to 0}\left[S_{F,a_{o}}^{\alpha }\left(x+dx\right)-S_{F,a_{o}}^{\alpha }\left(x\right)\right]\;if\;x,x+dx\in F\\ 0\;otherwise \end{array}\right.$$



**Theorem 3.** 
*The staircase function differential can be approximated by*

(13)
$$dS_{F,a_{o}}^{\alpha }\left(x\right)=\frac{A(\alpha )}{\alpha }dx^{\alpha }\;,$$

*where*

(14)
$$A\left(\alpha \right):=2\pi^{\alpha /2}/\Gamma \left(\alpha /2\right)\;.$$



**Proof.** For any volume ${\left(\delta x\right)}^{n}\in R^{n}$, its intersection with F has a volume ${\left(\delta x\right)}^{\alpha }$. Consider the volume of an *n*-dimensional sphere of radius *x* given by
(15)$$V\left(x\right)=\frac{A(n)}{n}x^{n}\;,$$
where $A\left(n\right)=2\pi^{n/2}/\Gamma \left(n/2\right)$ is the surface area term, with Γ(z) being the Euler’s Gamma Function, and $x=\sqrt{x_{1}^{2}+\cdots +x_{n}^{2}}$. Then, the volume of a spherical shell of finite width δx is given by
(16)$$\delta V\left(x\right)=\frac{A(n)}{n}\left({\left(x+\delta x\right)}^{n}-x^{n}\right)\;.$$□

In the limit δx→dx, where now dx is infinitesimal, it results

(17)$$dV\left(x\right)=A\left(n\right)x^{n-1}dx=\frac{A(n)}{n}dx^{n}\;,$$
where $dx^{n}:=d\left(x^{n}\right)$.

The intersection of δV(x) with F, which is denoted by $\delta V_{\alpha }\left(x\right)$, is
(18)$$\delta V_{\alpha }\left(x\right)=\frac{A(\alpha )}{\alpha }\left({\left(x+\delta x\right)}^{\alpha }-x^{\alpha }\right)\;.$$
In the limit δx→dx, this leads to
(19)$$\delta V_{\alpha }\left(x\right)\to dV_{\alpha }\left(x\right)=A\left(\alpha \right)x^{\alpha -1}dx=\frac{A(\alpha )}{\alpha }dx^{\alpha }\;.$$
On the other hand, $dS_{F,a_{o}}^{\alpha }\left(x\right)$ is the volume of the intersection between an infinitesimal volume $dV\in R^{n}$ with F. (The multiplicative coefficient A(α) used here is valid for integer dimensions. The case of fractional dimensions is more challenging, so this coefficient needs to be considered with care. In this work, we focus on the shape of the continuous approximation.)
(20)$$dS_{F,a_{o}}^{\alpha }\left(x\right)=\frac{A(\alpha )}{\alpha }dx^{\alpha }=A\left(\alpha \right)x^{\alpha -1}dx\;.$$

**Definition 10.** 
*The continuous approximation of a fractal function is defined as a set of infinitesimal elements dx such that Equation *(Equation 20)* is satisfied.*


**Theorem 4.** 
*The continuous approximation of the fractal derivative of a function is*

(21)
$$D_{F,\varphi }^{\alpha }h\left(x\right)=\frac{A(\alpha )}{\alpha }\frac{dh^{\alpha }}{dx}=A\left(\alpha \right)h^{\alpha -1}\left(x\right)\frac{dh}{dx}\left(x\right)\;.$$



**Theorem 5.** 
*The continuous approximation of the fractal derivative of a fractal function is*

(22)
$$D_{F,\varphi }^{\alpha }h\left(x\right)=\frac{A(\alpha )}{\alpha }\frac{dh^{\alpha }}{dx}=A\left(\alpha \right)h^{\alpha -1}\left(x\right)\frac{dh}{dx}\left(x\right)\;.$$



**Proof.** It follows directly from the definition of the fractal derivative of a function and of the continuous approximation. □

**Theorem 6.** 
*Consider a fractal function $f:R^{n}\to F$, where F is a $\delta^{\alpha }$-cover, with n−1<α<n, for n>1. It defines a set of fractal functions $\left\{f_{i}\left(x_{i}\right)\right\}$ with dimensions $\left\{\alpha_{i}\right\}$ such that $\alpha =\alpha_{1}+\cdots +\alpha_{n}$.*


**Proof.** Consider the fractal function $f_{k}\left(x_{k}\right)=f\left(a,\cdots ,x_{k},\cdots ,z\right)$, where a,⋯,z are constants. For any interval $I=[x_{k},x_{k}+\delta x_{k}]$, the intersection of *I* and F is ${\left(\delta x_{k}\right)}^{\alpha_{k}}$, with $\alpha_{k}<1$. For an $\alpha_{k-1}$-dimensional function $h_{k-1}\left(x_{1},\cdots ,x_{k-1},k,l,\cdots ,z\right)$ such that for any volume ${\left(\delta x\right)}^{k-1}$, the intersection with F is ${\left(\delta x\right)}^{\alpha_{k-1}}$, the function $h_{k}\left(x_{1},\cdots ,x_{k-1},x_{k},l,\cdots ,z\right)$ has dimension ${\left(\delta x\right)}^{\alpha_{k-1}}\delta x={\left(\delta x\right)}^{\alpha_{k}}$, where $\alpha_{k}=\alpha_{k-1}+\alpha_{k}$. The theorem is proved by induction. □

**Definition 11.** 
*Consider a fractal function h with dimension α<1. The gradient of a fractal function is defined as*

(23)
$$D_{F,\varphi }^{\alpha }h\left(x_{o}\right)=\left(D_{F,\varphi }^{\alpha_{1}}{{|}_{1}\;h\left(x_{o}\right),\cdots ,D_{F,\varphi }^{\alpha_{n}}|}_{n}\;h\left(x_{o}\right)\right)\;,$$

*where $\alpha =\alpha_{1}+\cdots +\alpha_{n}$.*


**Definition 12.** 
*For α>1, the partial fractal derivative of the function is*

(24)
$$D_{F,\varphi }^{\alpha }{|}_{i}h\left(x_{o}\right)=\left(D_{F,\varphi }^{\alpha_{1}}{{|}_{i}\;h\left(x_{o}\right),\cdots ,D_{F,\varphi }^{\alpha_{n}}|}_{i}\;h\left(x_{o}\right)\right)\;,$$

*where $\alpha =\alpha_{1}+\cdots +\alpha_{n}$.*


**Theorem 7.** 
*For a finite δ, the derivative of a fractal function in the interval [x−δ,x] is*

(25)
$$D_{[\delta ],\varphi }^{\alpha }h\left(x\right)=\frac{A(\alpha )}{\alpha }\int_{x-\delta }^{x}h^{\alpha -1}\left(t\right)\frac{dh}{dt}dt\;.$$



**Proof.** The derivative in the interval [x−δ,x] is
(26)$$D_{[\delta ],\varphi }^{\alpha }h\left(x\right)=\int_{x-\delta }^{x}D_{F,\varphi }^{\alpha }h\left(t\right)dt\;.$$□

Using Definition 10, the theorem is proved.

**Theorem 8.** 
*For a finite δ, the derivative of function in the interval [x−δ,x] in a fractal space is*

(27)
$$D_{[\delta ],a}^{\alpha }h\left(x\right)=\frac{A(\alpha )}{\alpha }\int_{x-\delta }^{x}{\left[h\left(x\right)-h\left(t\right)\right]}^{\alpha -1}\frac{dh}{dt}dt\;.$$



**Proof.** The proof is performed by applying the continuous approximation in Equation (Equation 20) to the derivative on fractal space in Definition 5. □

Observe that the α-dimensional sphere needs not to be centered at φ for the fractal derivative of a fractal function, or at *a* for the derivative on a fractal space. The point *x*, where the derivative is calculated, can be set as the center of the sphere.

**Definition 13.** 
*The continuous approximation of the derivative of a function on a fractal space, based on α-dimensional sphere centered at x is indicated by $D_{F,x}^{\alpha }h\left(x\right)$.*


**Theorem 9.** 
*The continuous approximation of the derivative of a function on a fractal space, $D_{F,x}^{\alpha }h\left(x\right)$ in the interval [x−δ,x], for finite δ, is given by*

(28)
$$D_{F,x}^{\alpha }h\left(x\right)=\frac{A(\alpha )}{\alpha }\int_{x-\delta }^{x}{\left(x-t\right)}^{1-\alpha }\frac{dh}{dt}dt\;,$$

*which is proportional to Caputo’s derivative.*


**Proof.** The local continuous approximation, considering that the radius of the spherical shell is x−t, is determined from Theorem 5 as
(29)$$D_{F,x}^{\alpha }h\left(t\right)=A\left(\alpha \right){\left(x-t\right)}^{1-\alpha }\frac{dh}{dx}\left(t\right)\;.$$□

Using Definition 13, one has
(30)$$D_{F}^{\alpha }h\left(x\right)=\int_{x-\delta }^{x}D_{F,x}^{\alpha }h\left(t\right)dt\;,$$
leading to the proof of the Theorem.

**Definition 14.** 
*The continuous approximation of the derivative of a fractal function based on α-dimensional sphere centered at x is indicated by $D_{F}^{\alpha }h\left(x\right)$.*


**Theorem 10.** *The continuous approximation of the derivative of a fractal function, $D_{F,\varphi_{x}}^{\alpha }h\left(x\right)$ in the interval [x−δ,x], for finite δ, is given by*(31)$$D_{F,\varphi_{x}}^{\alpha }h\left(x\right)=\frac{A(\alpha )}{\alpha }\int_{x-\delta }^{x}{\left(\varphi_{x}-h\left(t\right)\right)}^{\alpha -1}\frac{dh}{dt}dt\;,$$for *t* such that $h\left(t\right)<\varphi_{x}=h\left(x\right)$.

**Proof.** The proof follows the same lines of the proof for Theorem 9. □

**Corollary 3.** 
*The continuous approximation in Definition 10 is proportional to the limit of the continuous approximation in the range [x−δ,x] for δ→0 of Caputo’s derivative.*


## 3. Discussion and Conclusions

The fractal derivative proposed by Parvate and Gangal, presented in Definition 5, is the closest concept to the Hausdorff concept of fractional dimension spaces. Therefore, it is considered as the starting point for the analysis of fractal derivatives and fractional derivatives here.

The existence of the inverse of the Parvate–Gangal derivative is a natural consequence, i.e., a derivative of a function with a fractal image space that is defined on a domain space, which may or may not be fractal. This is proven in Theorem 1.

This work demonstrates that fractal functions with arbitrary dimension α, such as a fractal vector field with fractal dimension α>1, can be defined. However, the cases of most interest are those with α≤1, as they are physically relevant for the present work.

The derivative of a fractal function on a fractal space allows for a continuous approximation, as demonstrated in Theorem 4. Additionally, a similar continuous approximation can be obtained for the derivative of a function in a fractal space, as shown in Theorem 5. This approximation is identical to the *special derivative* used in Ref. [19] to derive the Plastino–Plastino Equation, which is a generalization of the Fokker–Planck Equation for systems with non-local correlations.

To illustrate the behavior of the continuous approximation, we utilize the well-known Cantor Set, which has a dimension α=0.631. We aim to demonstrate how the continuous approximation aligns with the mass distribution, $S_{F,0}^{\alpha }\left(x\right)$. To achieve this, we numerically calculate the mass distribution for this fractal set up to level 4. In other words, the smallest component of the fractal has a linear length of $l=3^{-4}$. We employ a δ-cover with δ=0.01 to calculate the mass distribution.

Next, we fit a power-law function, $y\left(x\right)=ax^{b}$. According to the theoretical findings presented in this work, the exponent *b* should closely approximate the fractal dimension α of the Cantor Set. The obtained results are displayed in Figure 1, revealing that the best fit corresponds to b=0.636, which is in close proximity to the expected value. This outcome effectively illustrates the application of the continuous approximation and provides insight into substituting the mass distribution by the continuous approximation. It should be noted that there are numerous other methods available for creating a continuous approximation of the fractal measure, and each of these approaches will result in different fractional derivatives. Investigating the coherence and convenience of different forms of approximation to the staircase function is an interesting line of research that is beyond the scope of this present study.

The continuous approximation derivative is expressed in terms of the standard derivative operator and can be associated with the *q*-deformed calculus [16]. Unlike the fractal derivative, the continuous approximation is a local derivative, and the non-linear behavior of the continuous approximation is a remnant of the non-local properties of the fractal derivative.

Non-locality can be explicitly introduced into the continuous approximation by considering finite δ-covers. In the non-local continuous approximation, the derivative is obtained by integrating the local continuous derivative over a finite range δ. This non-local continuous approximation is presented in Theorem 9, and it is precisely the Caputo fractional derivative.

The derivative of a function in a fractal space and the derivative of a fractal function lead to different continuous approximations. The former can be associated with the Caputo fractional derivative, as shown in Theorem 9, while the latter leads to a Caputo-like derivative, as demonstrated in Theorem 10. Similar derivatives to Caputo’s derivative can also be found in [20].

The results of the present work evidence the relations between the fractal derivative and some of the most used fractional derivatives. Comparing the result of Theorem 5 with Equation (Equation 4), it is clear that the local continuous approximation of the derivative of a fractal function is equal to the *q*-derivative. Thus, for the first time, the *q*-calculus derivative is shown to be a continuous approximation to the fractal derivative.

A consequence of the relationship between the *q*-derivative and the local continuous approximation of the derivative of a fractal function (Theorem 5), and of the connection between the derivative of a fractal function and the Caputo-like fractional derivative (Theorem 10) is that the *q*-derivative and the Caputo-like derivative are connected through a dislocation of the center of the α-dimensional sphere around which the non-local continuous approximation is calculated. Hereby, one can conclude that different forms of fractional derivatives can be obtained from the Parvate–Gangal fractal derivative by considering the different possibilities of continuous approximation and non-locality of the fractional derivative.

Other fractal derivatives can be explored along the same lines as performed here. The Riemann–Liouville derivative bears a close relationship with Caputo’s derivative [21] and it is interesting to observe the similarities between the fractal derivative proposed in Refs. [22,23] and the continuous approximations studied in the present work. The fractional derivative used in Ref. [24] is equal to the local continuous approximation of the fractal derivative of a function in a fractal space obtained in the present work. Ref. [25] studied this fractional derivative and its relationship with the q-derivative. Establishing a clear connection between the Parvate–Gangal fractal derivative and Caputo’s fractional derivative, this work opens the possibility for a deeper understanding of the use of fractional differential equations, which is so common in many different areas. In this respect, let us remark that fractal and fractional differential equations have been used in applications as dynamic of the system in porous or heterogeneous media [26,27,28], diffusive flow [29,30,31,32,33], solitons [34], control of complex systems [35], epidemic process [36], polymer plasma [37] and many others. The consequences of the present study for these physical systems deserve further investigation in the future. The consequences of the present study for these physical systems deserve further investigation in the future. Its implication on the study and applications of fractal functions [38] deserves further investigation.

## Figures and Tables

**Figure 1 entropy-25-01008-f001:**
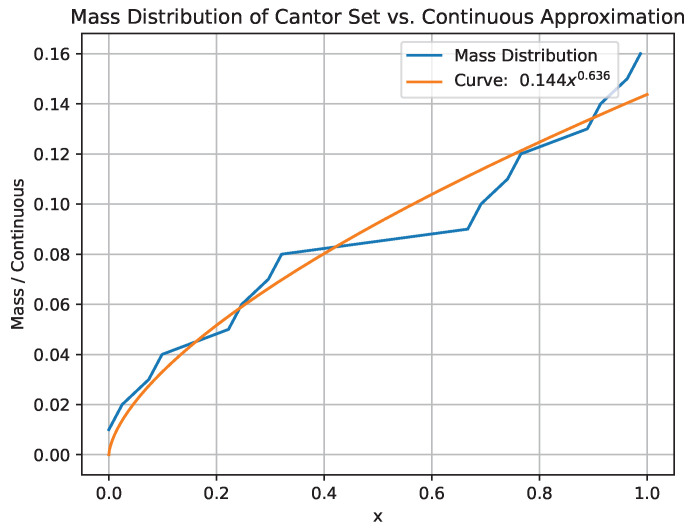
Plots of the mass distribution (blue line) for the Cantor Set at the 4th iteration, calculated with a δ-cover with δ=0.01, compared with the continuous approximation (orange line) represented by a function $y\left(x\right)=ax^{b}$ fitted to the mass distribution. The best-fit results in b=0.636, in agreement with the Cantor Set dimension α=0.631.

## Data Availability

Not applicable.

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
