# Peer review of "Fractal Derivatives, Fractional Derivatives and q-Deformed Calculus"

_entropy, 2023, doi:10.3390/e25071008_

Round 1

Reviewer 1 Report

The article under review looks like, on the one hand, a mini-review, and on the other hand, like a mini-introduction to such a section of analysis as fractal systems and fractional derivatives.

For the special issue `Non-additive Entropy Formulas: Motivations and Applications’, such an introduction is clearly necessary, since non-additive entropy is associated with non-local correlations, and non-locality is a key attribute of fractional derivatives.

The authors give definitions of the fractal derivative and establish the relationship between the fractional and fractal derivatives, as well as q-calculus. Fractal derivatives have been defined in mathematics relatively recently and are an important theoretical tool for the analysis of fractal systems. Applications of fractal derivatives go far beyond problems related to non-additive entropy.

The material of the article is presented in a compact, readable form. No significant errors were found in the text.

I recommend the article for publication in the special issue of `Non-additive Entropy Formulas: Motivations and Applications’ of `Entropy’ journal.

Author Response

The article under review looks like, on the one hand, a mini-review, and on the other hand, like a mini-introduction to such a section of analysis as fractal systems and fractional derivatives.

For the special issue `Non-additive Entropy Formulas: Motivations and Applications’, such an introduction is clearly necessary, since non-additive entropy is associated with non-local correlations, and non-locality is a key attribute of fractional derivatives.

The authors give definitions of the fractal derivative and establish the relationship between the fractional and fractal derivatives, as well as q-calculus. Fractal derivatives have been defined in mathematics relatively recently and are an important theoretical tool for the analysis of fractal systems. Applications of fractal derivatives go far beyond problems related to non-additive entropy.

The material of the article is presented in a compact, readable form. No significant errors were found in the text.

I recommend the article for publication in the special issue of `Non-additive Entropy Formulas: Motivations and Applications’ of `Entropy’ journal.

We thank the Reviewer for the attentive reading of the paper and for recommending the acceptance.

Reviewer 2 Report

1. The paper aims to develop some connections between q-calculus and fractional calculus. 

2. A lot of theorems are proved which looks fairly logical.

3. The claim of developing calculus for fractals has to be demonstrated with credible numerical simulations for wider acceptance.  

4. Ref 9-10 are same. Literature survey is not thorough on this topic. 

5. The paper is inaccessible unless utility of the mathematical results are shown on some real-world applications using credible numerical analysis and simulations.

Can be improved.

Author Response

1. The paper aims to develop some connections between q-calculus and fractional calculus.

2. A lot of theorems are proved which looks fairly logical.

We thank the Reviewer for her/his positive comments.

3. The claim of developing calculus for fractals has to be demonstrated with credible numerical simulations for wider acceptance.

We do not claim to have developed the fractal calculus, but we use the formalism developed by Parvate and Gangal (see refs 9 and 10). The original aspects of the present work is the discussion about the continuous approximation and its relation with q-deformed calculus. In the present version we provide numerical verification of the approximation. We are thankful for the suggestion given by the Reviewer, that improved the quality of the work allowing the reader to understand in depth the significance and the quality of the approximation. To clarify this point, we added the following sentence at the end of the Introduction:

In this work, we assume that the fractal derivative is correctly calculated by the definitions advanced by Parvate and Gangal [9,10], and discuss how some relevant forms of fractional derivatives, as well as the q-deformed derivative, can be obtained as a continuous approximation of the fractal derivative.

The numerical calculation is intended to show how the continuous approximation represents, the fractal structure contained in the mass distribution. The results show to what extend the fractal information is maintained in this approximation, and clarifies the significance of the approximation and, hereby, of the fractional derivatives associated with this approximation. The numerical calculation allowed us to observe that the surface term, A(alpha), is valid for integer dimensions, but may be not valid for fractional dimensions. To the best of our knowledge, the determination of the surface term for fractional dimensions is a problem that remains unsolved in fractal geometry. Therefore, we added a footnote informing that: “The multiplicative coefficient A(alpha) used here is valid for integer dimensions. The case of fractional dimensions is more challenging, so this coefficient needs to be considered with care. In this work, we focus on the shape of the continuous approximation.”.

4. Ref 9-10 are same. Literature survey is not thorough on this topic.

This error was corrected. We added four new references in the current version.

5. The paper is inaccessible unless utility of the mathematical results are shown on some real-world applications using credible numerical analysis and simulations.

We discuss some applications, including the study of dynamical aspects of thermofractals. A complete study of the dynamic of a fractal system is under development, but it is too large to be included in the present manuscript. We do include, however, numerical application for the continuous approximation, as described above.
